# SubjECTive-QA: Measuring Subjectivity in Earnings Call Transcripts' QA Through Six-Dimensional Feature Analysis

**Huzaifa Pardawala**[*][✉], **Siddhant Sukhani**[*][✉], **Agam Shah**[*][✉], **Veer Kejriwal, Abhishek Pillai,**
**Rohan Bhasin, Andrew DiBiasio, Tarun Mandapati, Dhruv Adha, Sudheer Chava**

Georgia Institute of Technology
✉ Corresponding Authors: {hpardawala3, ssukhani3, ashah482}@gatech.edu
* Indicates equal contribution

## Abstract

Fact-checking is extensively studied in the context of misinformation and disinformation, addressing objective inaccuracies. However, a softer form of misinformation involves responses that are factually correct but lack certain features such as clarity and relevance. This challenge is prevalent in formal Question-Answer (QA) settings such as press conferences in finance, politics, sports, and other domains, where subjective answers can obscure transparency. Despite this, there is a lack of manually annotated datasets for subjective features across multiple dimensions. To address this gap, we introduce SubjECTive-QA, a human annotated dataset on Earnings Call Transcripts' (ECTs) QA sessions as the answers given by company representatives are often open to subjective interpretations and scrutiny. The dataset includes $49,446$ annotations for long-form QA pairs across six features: `Assertive`, `Cautious`, `Optimistic`, `Specific`, `Clear`, and `Relevant`. These features are carefully selected to encompass the key attributes that reflect the tone of the answers provided during QA sessions across different domains. Our findings are that the best-performing Pre-trained Language Model (PLM), RoBERTa-base, has similar weighted F1 scores to Llama-3-70b-Chat on features with lower subjectivity, such as `Relevant` and `Clear`, with a mean difference of 2.17% in their weighted F1 scores. The models perform significantly better on features with higher subjectivity, such as `Specific` and `Assertive`, with a mean difference of 10.01% in their weighted F1 scores. Furthermore, testing SubjECTive-QA's generalizability using QAs from White House Press Briefings and Gaggles yields an average weighted F1 score of 65.97% using our best models for each feature, demonstrating broader applicability beyond the financial domain. SubjECTive-QA is publicly available under the CC BY 4.0 license[1].

## 1 Introduction

Earnings Calls (ECs) and their linguistic nuances serve as a vital communication channel between company executives and investors, offering insights into a company's performance and future outlook [Sawhney et al., 2021]. The long-form Question and Answer (QA) sessions of these calls are particularly significant as they provide unscripted interactions that reveal executives' confidence and strategic clarity. Unlike the scripted presentations in the beginning of ECs, the dynamic nature

---

[1]https://github.com/gtfintechlab/SubjECTive-QA

of QA sessions invites real-time scrutiny [Matsumoto et al., 2011] and deeper analysis [Alhamzeh et al., 2022] by analysts and investors as seen in figure 1. Linguistic nuances like tone and sentiment often predict abnormal returns more effectively than the actual earnings surprises disclosed [Price et al., 2012]. Traditional approaches to gauging business subjectivity often rely on indices which measures small business sentiment through survey responses that reflect managers perceptions and expectations about economic conditions. [Zorio-Grima and Merello, 2020]

Traditionally analyzed for financial insights, the dynamic nature of these QA interactions have broader applications across various domains including but not limited to presidential debates, journalism, and sports conferences, where the manner of information delivery is as critical as the content itself. Additionally, increasing amount of misinformation is not only about outright falsehoods but also about subtly misleading answers; these answers can be technically true but misleading or irrelevant, a challenge highlighted in a recent study by Li et al. [2021].

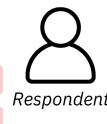

I **guess** you **mentioned** that there was some **interest**, is that in the **whole package** or **parts of it**?

*Questioner*

Its right, but we are in the middle of the process, so we really can't talk about specifically.

*Respondent*

❌ Relevant, Specific   ✅ Cautious, Assertive
❌ Clear, Optimistic   ❓ Soft Misinformation

Figure 1: An example of misinformation being present within question answer pairs of ECTs which is taken from the ECT of SWN in 2012 quarter 3.

The jargon-heavy nature of ECTs, often exceeding $5,000$ words, poses a complexity for retail investors [Koval et al., 2023]. The complexity and forward-looking statements within ECs underscore the need for specialized approaches in Financial Natural Language Processing (FinNLP) to effectively handle and interpret this voluminous and nuanced information. Existing FinNLP datasets derived from ECT data predominantly focus on sentiment classification, stock price prediction [Medya et al., 2022], summarization [Mukherjee et al., 2022], and objective annotation of financial statements, overlooking the subjective nuances embedded within the QA exchanges.

Recognizing these gaps, our paper introduces SubjECTive-QA, a pioneering dataset that enriches the financia domain. This dataset provides subjectively annotated responses from EC long-form QA sessions, with an average QA length of nearly 186 words, covering 120 ECTs of companies listed on the New York Stock Exchange from 2007 to 2021. Unlike traditional datasets that either quantify sentiment or dissect financial statements into objectively verifiable claims [Maia et al., 2018], SubjECTive-QA delves into the multifaceted nature of answers, offering a novel lens through which financial discourse can be evaluated. The meticulous annotation of these transcripts with a six-

Table 1: Overview of the SubjECTive-QA dataset.

| Metric | Value |
| --- | --- |
| Dataset size | $35,711$ |
| Total QA Pairs | $2,747$ |
| Total Features | 6 |
| Total Metadata columns | 7 |
| Total Annotations | $49,446$ |
| Avg. Question Length | 59.87 words |
| Avg. Answer Length | 127.15 words |
| Unique Questioners | 756 |
| Unique Responders | 305 |

label subjective feature rating system aids in capturing the dimensions of clarity, assertiveness, cautiousness, optimism, specificity, and relevance. Our aim is to provide a comprehensive resource that transcends traditional sentiment analysis. The statistical details of SubjECTive-QA are illuminated in table 1.

Furthermore, our dataset and methodology extend beyond the financial domain, addressing the need for robust subjectivity and misinformation detection tools applicable in various domains such as elections, journalism, sports, and public policy. QA sessions are prevalent in these areas, where the quality and clarity of responses significantly impact decision-making and public perception. As shown in Appendix N, we applied our models to White House Press Briefings [The White House, 2024], a setting where transparency and caution are paramount. Our analysis of QA pairs from White House Press Briefings and Gaggles demonstrates the utility of our models in a political con-

text. These findings underscore SubjECTive-QA's effectiveness in capturing the nuanced subjective information required in such high-stakes environments. As we delve deeper into the application and evaluation of various Natural Language Processing (NLP) models on the SubjECTive-QA dataset, it is imperative to understand how these models perform in capturing the specific features identified.

In our benchmarking efforts, various NLP models are evaluated on the SubjECTive-QA dataset to measure their effectiveness in capturing these features. While general-purpose models like BERT base (uncased) [Devlin et al., 2018] and RoBERTa-base [Liu et al., 2019] perform well, the results underscore the importance of domain-specific models like FinBERT-tone [Huang et al., 2020] which shows higher accuracy in certain features. This work hence not only advances FinNLP but also sets a precedent for broader applications in detecting subjectivity and misinformation in diverse QA contexts.

We aim to contribute significantly to the fields of QA session analysis and NLP by enabling researchers and practitioners to use our dataset as a valuable resource to assess the quality of information across multiple domains.

## 2  Features in SubjECTive-QA

SubjECTive-QA consists of six features for analyzing the quality of speech of the respondent. These features and their definitions are given in table 2. This process was initialised with an LLM-guided approach: passing each QA pair to the PaLM 2 API [Anil et al., 2023], to obtain the 10 most prevalent properties demonstrated by the answer for that particular question. This approach is elaborated in detail in Appendix I.

Table 2: Feature descriptions utilized within SubjECTive-QA, explaining the definitions used for annotation purposes as well as the reason for choosing these features.

| Feature | Description | Justification of choice |
| --- | --- | --- |
| Relevant | The speaker has answered the question with appropriate details. | In a formal environment such as during an EC, relevant answers indicate the speaker addresses concerns directly. Irrelevant answers would lead to poor communication and potential misunderstandings about the company's strategy or performance. |
| Clear | The speaker is transparent in the answer and about the message to be conveyed. | Clarity is crucial in formal environments. It ensures that the speaker's message is well understood and transparent, which is often expected in environments like an EC. |
| Optimistic | The speaker answers with a positive outlook regarding future outcomes. | Optimism signals expectations of better future results and performance, indicating the company anticipates favorable tailwinds. |
| Specific | The speaker includes sufficient and technical details in the answer. | Specific answers demonstrate technical and statistical accuracy, which is important in ensuring transparency and reliability in an EC. |
| Cautious | The speaker answers using a more conservative, risk-averse approach. | Cautiousness can indicate defensiveness or a lack of conviction in the company's future, but it may also reflect a prudent, risk-averse mentality. |
| Assertive | The speaker answers with certainty about the company's events and outcomes. | Assertiveness shows the competence and reliability of the speaker and the firm. High assertiveness can demonstrate persuasive abilities and trustworthiness. |

The final 6 chosen features were also seen to have a significant impact on almost all QA pairs as per visual inspection over the corpus of our data and the reason for choosing each specific feature can be seen in 2. All the features were shown to be independent after annotation as seen in figure 4. This independence criterion is paramount as it ensures that potential classifiers could be fine-tuned to focus on any one of the 6 features without having to account for the others.

# 3   Methodology

The creation of SubjECTive-QA is depicted below in figure 2. Table 1 details the metrics of our dataset captured by this process.

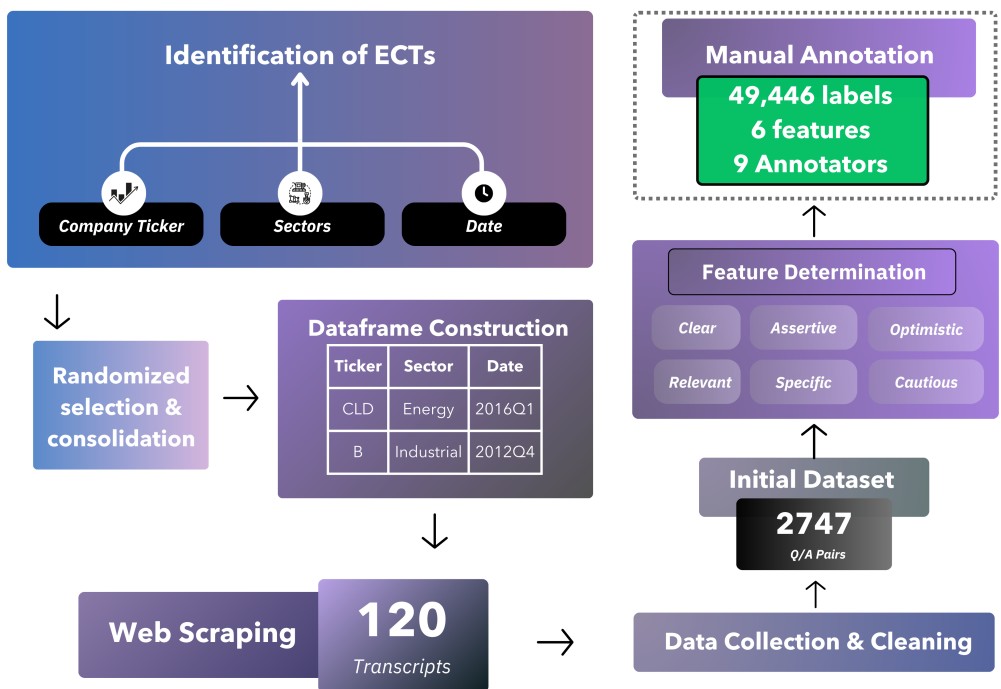

Figure 2: Compact overview of the dataset construction process utilized when constructing SubjECTive-QA.

## 3.1   Dataset Construction

**Identification and Selection**   We commenced with the identification of company tickers, sectors, and verification of EC dates from 2007 to 2021. A foundational dataset obtained from Chava et al. [2022] facilitated the initial sampling and offered crucial metadata.

The variables from this dataset that were utilized include company name, EC date (for year and quarter selection), and sector the company operate in. This metadata enabled the precise identification of ECs for our subsequent data collection.

**Sampling and Data Collection**   We then proceeded with the randomized selection of companies and their respective earnings call dates from the $119,978$ records obtained from Chava et al. [2022], weighing each choice by an even time distribution and sectors already sampled from. This stage was critical for ensuring a diverse and representative dataset, unaffected by biases towards the specific sectors or time periods. Utilizing a combination of the previously highlighted selectors, our algorithm methodically chose 120 earnings calls from the list of company tickers. This procedure is outlined extensively in Appendix H.2.

**Sector Consolidation**   When consolidating the data, there were significant overlaps in QA structures and linguistic patterns in the ECTs across the original sectors in terms of the features chosen. To address this, the 13 unique columns in the original dataset as mentioned in Appendix H.1 were reclassified and mapped to unique numerical values based on the similarities we found to be present within their ECs. This mapping can be seen in the table 7 in Appendix H.1 with the industries' respective numerical labels. These measures resulted in the construction of a more comprehensive

dataset with new industry labels. However, this dataset still contained a significant amount of noise in terms of verbal-filler QA pairs such as salutations, formal introductions and irrelevant text as illustrated with examples in table 8 in Appendix H.1.

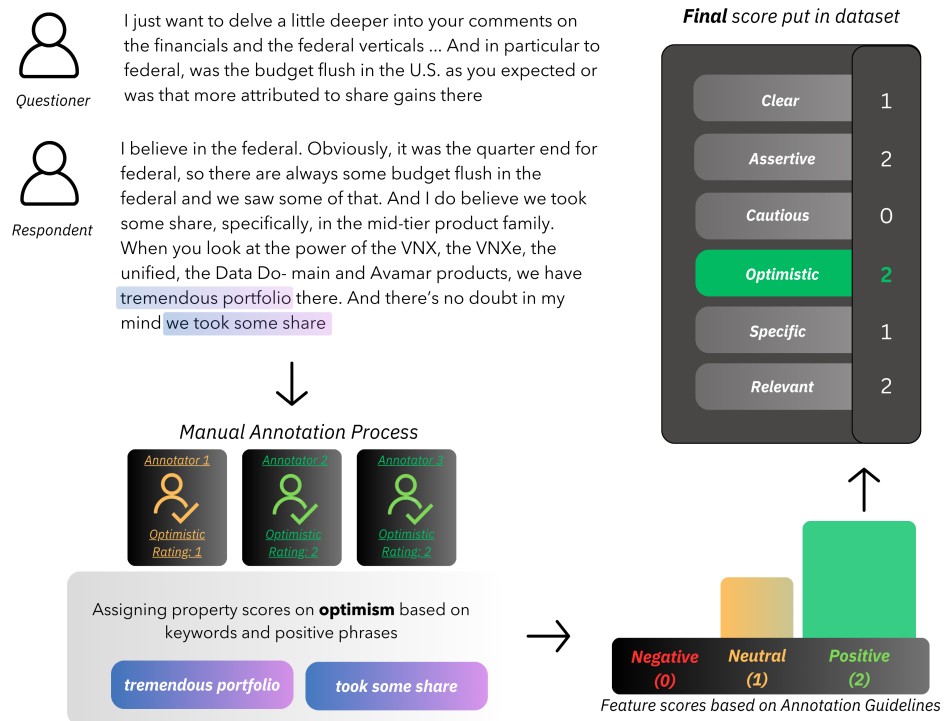

Figure 3: An example of the annotation process used while generating a rating for the `Optimistic` feature, indicating the reasons for choosing 2 as the rating.

**Data Cleaning**   In order to remove verbal filler content within our dataset without losing valuable data, we employed a manual cleaning process. This involved the authors going through each collected QA record to filter the data to remove filler content. Additionally, in case a questioner asked a question and there were multiple respondents who answered the questions subsequently, each of these answers and the respondents were mapped to the same question and questioner, establishing new, individual records. An example of the data cleaning process can be found in table 8 in Appendix H.1. After cleaning up the data, we began the manual annotation phase with our team of annotators.

## 3.2   Annotations

**Annotators**   The last step was the manual annotation of each QA pair across the 6 features mentioned in 11. Each ECT was randomly assigned to three annotators. For each ECT, the annotators remained anonymous to one another. The team of annotators comprised nine people whose details are outlined in table 12 in Appendix M.1.

**Annotation Guideline**   This paper employed Microsoft Excel for the annotation procedure and figure 3 illustrates the manual annotation process. The annotators were asked to strictly adhere to the following annotation guidelines:
Give the answer a rating of:

- 2: If the answer positively demonstrates the chosen feature, with regards to the question.
- 1: If there is no evident relation between the question and the answer for the feature.

- 0: If the answer negatively demonstrates the chosen feature with regards to the question.

At the end, the individual annotations were combined based on majority rating. In case there was no clear majority that particular rating was assigned the value '1'. A sample annotation is shown in Appendix G to make the annotation procedure clearer. We highlight several QA pairs in table 3 and detailed annotations for each QA pair in table 4 as a sample for our annotation work in Appendix G. The ethical considerations for our annotations are outlined in Appendix F.

**Annotator Agreement** The annotator agreement metrics were calculated by obtaining the percentage of times the annotators completely agreed (all 3 annotators agree on the same rating), partially agreed (2 of the annotators agree and 1 disagrees) and completely disagreed (all 3 annotators had different ratings). We obtained an aggregate percentage for the annotator agreement scores across all 6 features. $48.94\%$ of the times the annotators completely agreed on a rating whereas $45.18\%$ one of the annotators disagreed with the other two. Lastly, all 3 annotators disagreed only $5.88\%$ of the times. The exact numbers for each feature are elaborated upon in Appendix M.2.

## 4 Dataset Analysis

### 4.1 Independence Criterion

Upon looking at the correlation matrix in figure 4, a general independence of features can be seen with the range of the correlations being between $-0.08$ and $0.39$. As stated before and verified through this correlation matrix, no significant relationship exists between any of the features, indicating that the chosen features uniquely classify the behaviour of the respondent and therefore can be independently modelled in the future. It is important to note that this correlation matrix disregards sector-wise bias between different features.

### 4.2 Rating Distribution

In order to measure the sector-wise bias and variable distributions of features across sectors, we utilized violin plots as seen in Appendix J as they allow for a compact representation of the kernel density and distribution of the data. Revealing specific asymmetries and skewness in various features across industries, these plots aided in the identification of specific behaviors and distributions of features across sectors.

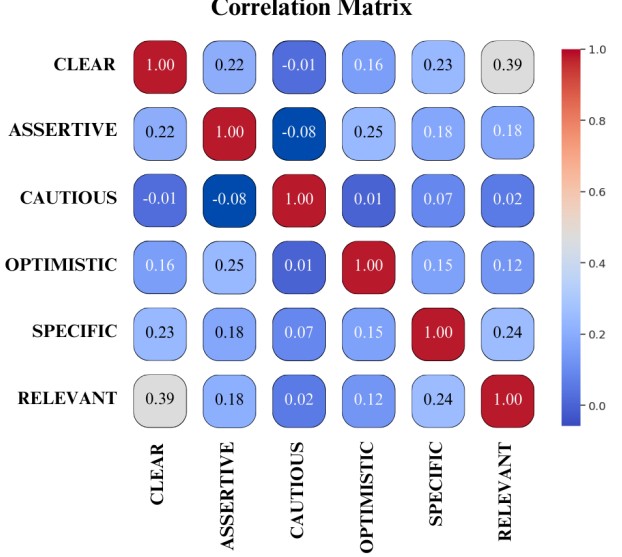

Figure 4: A correlation matrix depicting the general independence of features utilized within SubjECTive-QA using pearson correlation.

When considering the spread of the ratings across the features, it can be seen that around $90\%$ of answers were given a rating of 2 for `Clear` and `Relevant`, showing that most respondents answer questions in a cohesive manner that is contextually relevant. For answers with a rating of 0, the feature that had the highest number of zeroes was `Specific` with around a fifth of all QA pairs negatively demonstrating this feature, indicating the variance in the quality of answers as shown by its violin plots within Appendix J.

Further exploration into the distribution of the features `Clear` and `Relevant` supports the hypothesis that company representatives aim to be confident and on-topic with their violin plots being highly dense to the rating of 2; however, there is high variation in the `Specific` feature across industries,

suggesting a lack of technical details possibly to simplify information for a broader audience or to protect the company's reputation.

On the other hand, the violin plots demonstrate the telecommunications sector to be highly `Optimistic` with overwhelming positive responses and an overall buoyant industry sentiment. However, the respondents within this industry were highly `Cautious` as seen in its violin plots and this is apparent within all industries, displaying the conservative nature of the answers. Overall, a wide spectrum of densities across different features and industries demonstrate diversity in tones and attitudes, emphasising the multilayered complexity of the proposed dataset.

# 5  Benchmarking

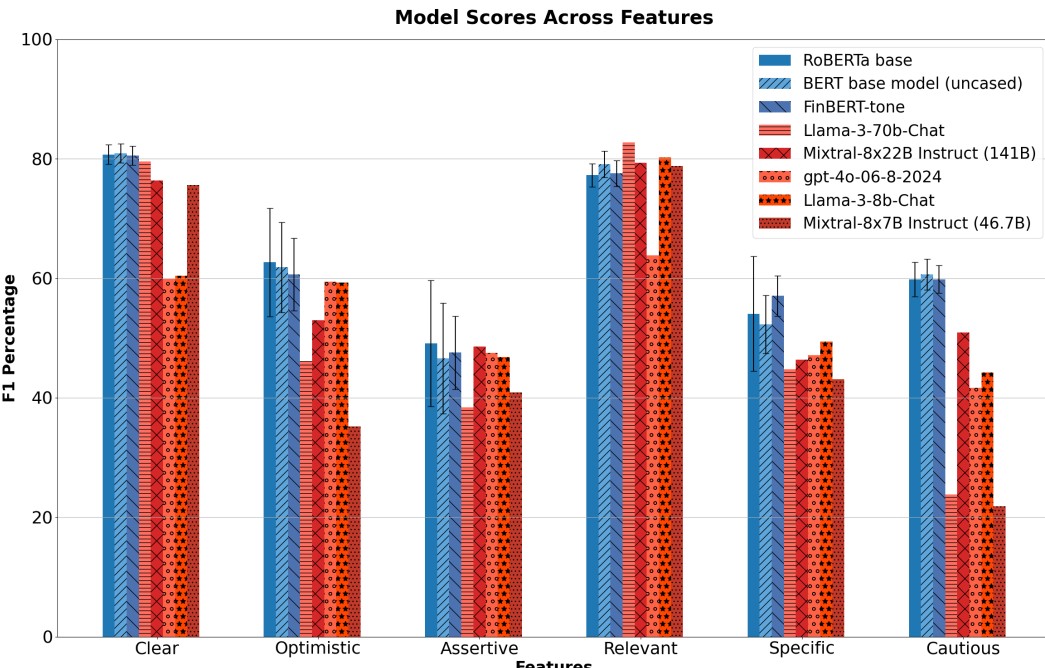

Figure 5: F1 percentage scores across several LLMs (red) and PLMs (blue) trained on SubjECTive-QA across all features as well as the error bars for the PLMs.

## 5.1  Models

**Pre-trained Language Models (PLMs)**  To establish a performance benchmark, our study encompasses a range of transformer-based Pre-trained Language Models. We employ BERT base (uncased), FinBERT-tone [Huang et al., 2020], and RoBERTa-base. To avoid overfitting on financial text data, we refrain from pre-training any of the models before fine-tuning them. The task performed is sequence classification, minimizing cross-entropy loss. The experiments are conducted using PyTorch [Paszke et al., 2019] on an NVIDIA A40 GPU. Each model is initialized with the pre-trained version from the Transformers library provided by Huggingface [Wolf et al., 2020]. We use varying hyperparameters and conduct multiple runs for each model using three seeds $(5768, 78516, 944601)$, three batch sizes $(32, 16, 8)$, and three learning rates $(1e-4, 1e-5, 1e-6)$. Following this, we utilize a grid search strategy to find the best model for each feature. The ethical considerations while using these models are outlined in Appendix F.

**Large Language Models (LLMs)**  Our study also encompasses four popular open-source LLMs: Llama-3-70b-Chat [Dubey et al., 2024], LLama-3-8b-Chat [Dubey et al., 2024], Mixtral-8x22B Instruct (141B)[Jiang et al., 2024], and Mixtral-8x7B Instruct (46.7B), and one closed-source LLM: GPT-4o-06-08-204 [OpenAI et al., 2024]. The hyperparameters for these models were as follows: `max_tokens`: 512, `temperature`: 0.0, `repetition_penalty`: 1.1. To access these models and

run the fine tuning code, we utilised together.ai API and we are thankful to them for providing us with free credits for the same. The ethical considerations while using these models are outlined in Appendix F.

## 5.2 Results

As seen in Figure 5, all models had similar performance on the dataset. While `Clear` and `Relevant` features were identified correctly a larger proportion of the time, the models' evaluation of `Assertive` and `Specific` were not as accurate. For each feature, we observed different models performing better. Due to the independence of our features, we can use each model independently to evaluate a given feature. For `Clear`, BERT had the highest weighted F1 score of 80.93%. For `Optimistic` and `Assertive`, RoBERTa-base had the highest weighted F1 scores of 62.69% and 49.10%, respectively. For `Relevant`, the LLMs, Llama-3-70b-Chat and Mixtral-8x22B Instruct (141B), outperformed the Pre-trained Language Models (PLMs), with Llama-3-70b-Chat achieving the highest weighted F1 score of 82.75%. For `Specific`, FinBERT had the highest weighted F1 score. This can be attributed to the fact that the other models are general-purpose models, whereas FinBERT is a domain-specific model for finance. For `Cautious`, BERT outperformed the other models with a weighted F1 score of 60.66%. Across all six features, RoBERTa-base had the highest average weighted F1 score of 63.95%. Mixtral-8x22B Instruct (141B) had a higher average weighted F1 score than Llama-3-70b-Chat.

The features Clear and Relevant were the easiest for models to identify, with BERT achieving the highest weighted F1 score of 80.93% for Clear and Llama-3-70b-Chat scoring 82.75% for Relevant. These features are more straightforward to detect as they rely on linguistic cues like coherence and topic alignment, making them accessible for general-purpose models. In contrast, detecting Assertive and Specific was more challenging. RoBERTa-base led in detecting Assertive with a score of 49.10%, while FinBERT excelled in identifying Specific, which relies on domain-specific technical details. These lower scores reflect the difficulty models face in capturing nuanced aspects such as tone and technicality.

**Analysis of Model Performance** FinBERTs higher performance for Specific emphasizes the value of domain-specific pre-training, as general-purpose models struggled with specialized financial terminology. The better performance on Clear and Relevant stems from their objective nature, as they rely on straightforward criteriawhether an answer is understandable and relevant to the question. However, detecting Assertive and Specific requires models to interpret subtle cues, making them harder to identify.

The performance discrepancies highlighted in this analysis open up important avenues for further research and development of models capable of handling subjective features more effectively. The challenges faced by current models in identifying nuances like assertiveness, cautiousness, and specificity suggest that standard pre-training on large corpora may not be sufficient for capturing complex human communication in high-stakes environments. Additionally, constructing richer training datasets with more nuanced annotation guidelines could help models learn to distinguish subtle variations in tone, sentiment, and technical specificity. This opens up opportunities to explore new architectures or techniques, such as reinforcement learning or attention mechanisms, that focus on capturing the intent and subjectivity behind language, thereby enhancing the models capacity to perform well in complex, subjective question and answer scenarios. Furthermore, a comparison of model latency, as explored by Shah and Chava [2023], could be an interesting direction for future work to assess the trade-offs between performance and efficiency.

## 5.3 Transfer Learning Ablations

This study evaluates the transfer learning capabilities of the best-performing model, RoBERTa-base, originally trained and tested on the SubjECTive-QA dataset. Specifically, we investigate its performance when fine-tuned on the SubjECTive-QA dataset, followed by testing on 65 question-answer pairs from White House Press Briefings and Gaggles as outlined in Section N with the outcomes of these transfer learning experiments. The model achieved a mean weighted F1 score of 65.97% across all the features, performing the best on `Clear` and the worst on `Cautious`. All the individual features' weighted F1 scores are outlined in Appendix N. This shows the broader applicability of

the dataset across different significant domains such as Politics where clarity and transparency are of utmost importance.

## 6 Related Works

**Subjective Datasets** Recent advancements in sentiment analysis such as Sy et al. [2023] have led to tailored tools and the creation of subjective datasets that have high potential within the financial domain. Many studies emphasize the importance of emotional information [Chen et al., 2023a] and linguistic extremity [Bochkay et al., 2020] on stock returns and investor opinions. The FinArg dataset curated by Alhamzeh et al. [2022] delves into argumentative sentiment while the General Numeral Attachment dataset generated by Shi et al. [2023] enhances numeral interpretation in ECTs, improving volatility forecasting. Similarly, Hiray et al. [2024] introduce CoCoHD, a dataset of U.S. congressional hearings, enabling sentiment and policy analysis on socio-economic issues, which complements our focus on multidimensional subjectivity in financial contexts. However, these datasets remain focused on a singular field, limiting their applicability across financial tasks. To optimize model performance, diverse data is crucial [Liang, 2016, Shah et al., 2022]. While the mentioned datasets take a unidimensional approach, our method leverages the multidimensional nature of ECTs to better capture sentiment.

**Sentiment Analysis and Annotations** Previous studies on the role of language in corporate reporting only take into account the tone of negative or positive words [TETLOCK, 2007, Loughran and McDonald, 2010] whilst our dataset focuses on a multidimensional analysis of 6 features. Most prior datasets also annotate single turn QA systems [Zhu et al., 2021, Qu et al., 2019, Li et al., 2022] without taking account the context of the question being asked [Deng et al., 2022]. Furthermore general sentiment datasets such as Malo et al. [2014] and Sinha and Khandait [2020] lose accuracy because they annotate over large text [Tang et al., 2023]. Our dataset aims to utilize the context of both questions and answers to augment our manual annotation process and incorporate a more nuanced annotation style to not lose accuracy.

**Earnings Calls** Based on their availability and the vast amount of information prevalent within them, ECTs proved to be a viable data source for our research. ECTs, hosted by publicly traded companies to discuss aspects of their earnings reports [Givoly and Lakonishok, 1980, Keith and Stent, 2019], remain to be a major form of communication that help investors to review their price targets and trade decisions [Frankel et al., 1999, Kimbrough, 2005, Matsumoto et al., 2011]. Recently, Shah et al. [2024] proposed a novel framework for fine-tuning LLMs on earnings call transcripts, integrating both sentiment and financial performance features, a method that enhances predictive power for earnings surprises. Secondly, sentiment analysis within ECTs has historically been proven to possess a correlation to earning surprises [Price et al., 2012, Bowen et al., 2002, Doyle et al., 2012], providing quantitative value of analyzing subjectivity in ECTs. Finally, sentiment analysis, fine tuning of LLMs, and deep learning tactics on ECTs offer valuable insight to predict companies future earnings surprises [Koval et al., 2023, Larcker and Zakolyukina, 2012] and emotional reaction [Bochkay et al., 2020, Chava et al., 2022] with reasonable accuracy. Our dataset allows for NLP models to be fine tuned on the subjectivity of ECTs with the goal to be generally used on QA pairs in various fields of research.

**Financial Domain QA Datasets** Appendix K provides a brief comparison of SubjECTive-QA with other datasets in the financial domain, focusing on the following attributes: size, number of features, list of labels, and license used. The datasets include TAT-QA [Zhu et al., 2021], a question-answering benchmark based on a hybrid of tabular and textual content in finance; FinQA [Chen et al., 2022a], a dataset designed for numerical reasoning over financial data; FinArg [Alhamzeh et al., 2022], which annotates argument structures in earnings calls; TruthfulQA [Lin et al., 2022], which measures how models mimic human falsehoods; Trillion Dollar Words [Shah et al., 2023], which evaluates the meeting minute sentences of the federal reserve of the United States; ConvFinQA [Chen et al., 2022b], which explores chains of numerical reasoning in conversational financial question answering; and MathQA [Amini et al., 2019], a dataset focused on interpretable math word problems.

# 7 Limitations and Future Work

**Earnings Calls Sampled**   Our dataset of Earnings Calls only encompasses companies listed in the New York Stock Exchange from 2007 to 2021 balanced across the 6 major industries defined in Appendix H.1. Insights from this dataset may not be applicable for Earnings Calls of companies from other countries or years. We plan to extend our research to other years and countries and test the broader applicability of our models.

**Manual Annotations**   The dataset of manual annotations was curated by the authors provided in the table 12 in Appendix M.1. As these annotations are subjective by definition, the dataset reflects a specific viewpoint and degree of financial knowledge given by the annotators' backgrounds. However, the subjectivity of the annotations presents itself as an interesting area for future work: analyzing perception of the subjective features in communication.

**Written Transcripts**   Our work uses written transcripts of ECs rather than the original audio. As a result, some aspects such as pitch, intonation, and tone that may be clear in an audio extract will not be reflected in the presented dataset [Sawhney et al., 2020]. The feature annotations may not demonstrate the same insights that an investor would discern through listening to an EC audio recording.

# 8 Discussion

SubjECTive-QA offers the first dataset of long form QA pairs annotated across six features. The dataset consists of $2,747$ QA pairs taken from 120 Earnings Call Transcripts annotated on six features: Clear, Assertive, Cautious, Optimistic, Specific, and Relevant. The goal of SubjECTive-QA is to serve as a resource for further research into the intersection between language and financial markets. Rather than solely focusing on the quantitative information within the Earnings Calls, measuring the various features present within QA pairs provides another dimension to analyze the effect of ECs on market dynamics. This paper defines the creation of SubjECTive-QA and examines introductory analysis into the distribution of our manual annotations. We believe that SubjECTive-QA can be a valuable resource for further exploration into the impact of ECs on financial markets and the FinNLP domain at large.

**Broader Impact:**   By capturing the intricate nuances of speech, our subjective dataset also lays the foundation for a new approach to identifying disinformation and misinformation. Conventional detection methods often fail to recognize its subtle linguistic cues, so our findings will prove vital. SubjECTive-QA therefore has applications beyond the field of FinNLP in various fields such as sports, news and politics to identify misinformation and disinformation. Through systematic analysis and refinement of the specifics of this dataset, researchers can develop algorithms capable of discerning various forms of disinformation, thereby advancing the field's ability to combat deceptive narratives effectively.

## Acknowledgments and Disclosure of Funding

We have not received any specific funding for this work. We appreciate the generous infrastructure support provided by Georgia Techs Office of Information Technology, especially Robert Griffin. We would like to thank Chandrasekaran Maruthaiyannan for his help with the annotations. We would also like to especially thank Michael Galarnyk for his valuable feedback and reviews. Furthermore, we greatly appreciate all the feedback from the reviewers which has helped us improve the paper and add some additional information for readers.

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

# Contents

## A Github and Hugging Face

## B Glossary

- **Soft misinformation** - Information that, while factually accurate, is presented in a manner that is ambiguous, irrelevant, or obscures the underlying truth.

- **Abnormal returns** - The difference between the actual return of a security and its expected return, generally used to assess the financial impact of specific events.

- **Sentiment** - An attitude, feeling, or opinion expressed in communication, often classified as positive, negative, or neutral. In data analysis, sentiment refers to the inferred emotional tone within a text, speech, or other form of media, which can indicate public opinion, mood, or general response toward a subject.

- **Ticker** - A unique series of letters assigned to a publicly traded company, used as its symbol on stock exchanges for identification. Tickers represent the company in trading and financial markets, enabling investors and analysts to quickly recognize and access information about the companys stock.

## C Abbreviations

| Abbreviation | Definition |
|---|---|
| NLP | Natural Language Processing |
| LLM | Large Language Model |
| PLM | Pretrained Language Model |
| ECT | Earnings Call Transcript |
| EC | Earnings Call |
| QA | Question-Answer |
| FinNLP | Financial Natural Language Processing |

## D Author Statement

The authors hereby confirm that we bear all responsibility in case of any violation of rights, including but not limited to intellectual property rights, privacy rights, and data protection regulations, that may arise from the use of the provided data. Furthermore, we confirm that the dataset SubjECTive-QA is licensed under the Creative Commons Attribution 4.0 International (CC BY 4.0) license, which allows others to share, copy, distribute, and transmit the work, as well as to adapt the work, provided that appropriate credit is given, a link to the license is provided, and any changes made are indicated.

## E Hosting, Licensing and Maintainence

The SubjECTive-QA dataset is available under the Creative Commons Attribution 4.0 International (CC BY 4.0) license, allowing users to share, adapt, and build upon the dataset, provided appropriate credit is given. Hosting for the SubjECTive-QA dataset is provided on both GitHub, offering versatile access options for researchers and developers. GitHub serves as a reliable platform for version control and collaborative contributions. The dataset is provided as-is and will not receive updates, ensuring a stable and consistent resource for users.

# F   Ethical Consideration

The work done in this research adheres to all ethical considerations and we do not identify any risks prevalent in the research conducted. However, we do acknowledge the presence of certain limitations and biases present in our research work due to educational, geographic and gender biases that are present within our annotation and research work.

- **Educational Bias** All researchers share a similar educational background and specialise in STEM based fields. This may have impacted the annotation process.

- **Demographic bias** 8 of the researchers are of Indian origin and 4 were born and brought from the same city within India. All researchers were present within the United States of America at time of writing and annotating the research work. The socioeconomic conditions and environment may have generated a bias in their work.

- **Geographic Bias** Our study focuses entirely on publicly listed companies within the United States of America, introducing a bias in the final annotated dataset. SubjECTive-QA hence may not be representative of ECTs of global markets and companies.

- **Gender Bias** There is a gender bias present within our study as the representatives and the analysts within the ECTs were predominantly male. Additionally, all the annotators were also male.

- **Data Ethics** Data collection will strictly adhere to the terms of service, legal regulations, and ethical guidelines governing publicly accessible sources. It should also be noted that all sources referenced are publicly accessible.

- **Annotation Ethics** The annotation of the dataset was completed by the authors of this paper, preventing any ethical concerns regarding the annotation process. None of the authors were paid to do the annotations.

- **Publicly Available Data** SubjECTive-QA will be made publicly available and we will also indicate the licenses under which it may be shared.

- **Language Model Ethics** The language models utilised in our research are publicly available, open source and fall under license categories that allow their usage for our intended purposes. The models used are cited and we acknowledge the environmental impacts of large language models and thus limit our work to fine-tuning pre-existing models.

- **Hyperparameter Reporting** All the hyperparameters utilised for training the models are specified within section 5. Our model setup is thus transparent and readers can get detailed information on how we trained our models.

- **PLM Ethics** Responsible AI practices will guide the utilization of Pre-trained Language Models.

- **LLM Ethics** Responsible AI practices will guide the utilization of the Large Language Models.

The research team is dedicated to promoting accessibility, fairness, and transparency by communicating any limitations in the research findings to ensure ethical integrity and promote responsible research practices.

# G   Sample Annotations

Table 3: Examples of various question-answer pairs taken from ECTs that will be used to detail the sample annotation mentioned in Table 4

| Sample | Question | Answer |
|---|---|---|
| A | I also have a couple of questions. I'm going to start with any comments you might offer about the ammunition supply chain, it's been something that's come up on recent calls. Do we take it from the fact you didn't comment on it that it's gotten a lot better? | Well, I think there's still some issues in terms of the supply chain. I think it has gotten better, but we're still not receiving all the ammunition, all the calibers that we would like to see in the quantities we would – ideally our consumers would like to see them. |
| B | I just want to delve a little deeper into your comments on the financials and the federal verticals, which doesn't sound like you had issues that others have been seeing. Does your commentary about the linearity of the business also apply to these verticals? And in particular to federal, was the budget flush in the U.S. as you expected or was that more attributed to share gains there? | I believe in the federal. Obviously, it was the quarter end for federal, so there are always some budget flush in the federal and we saw some of that. And I do believe we took some share, specifically, in the mid-tier product family. When you look at the power of the VNX, the VNXe, the unified, the Data Domain and Avamar products, we have tremendous portfolio there. And there's no doubt in my mind we took some share. |
| C | Can you just talk a little bit about the geographies? Joe, what did you see in Europe? It looks like pre-currency growth was probably like up around 10%. That's still actually pretty good for that environment. And APJ, why was it so good? | No, APJ is just really hot for us, as our other parts of the world ... So our revenues are still – and, of course, Q3 is probably more U.S. skewed which is 54% of revenues, but kind of in a – most of the other quarters you'll see our – we still have 52-ish percent of our revenues in the U.S. ... So we're very – so part of it is focused, part of it is the, I think, just the attractiveness of our product line. Part of it is the fact that we have younger and have a lot more growth opportunity relative to our potential outside the U.S., and I think that's why Asia. In Europe, as you said, as David said, in constant currency, we grew 12% which we think is pretty good... |
| D | Thank you so much. So, Peter, I think you wanted to kind of talk about potential permanent changes to consumer behavior. I think vacation rentals versus a hotel are fairly well understood. but are you noticing any change in terms of folks favoring agency versus merchant because I'm sure they probably learned last year they're paying ahead of time and trying to get refunds later on? It's probably something that they probably don't want to do again. So, are there kind of sort of meaningful differences in the conversion rate between the 2 types of transactions you can call out? And does that positively or negatively influence your customer acquisition strategies going forward? Thanks. | Yeah. And I will just add, Stephen, that we're not trying to drive, as Eric said, we're not trying to drive the customer to any particular outcome. We provide choices by and large. And what the customers do, what they want. There has been a relative bias during COVID for pay later. As you say perhaps, I could later do something security around the idea. But there's nothing that I think suggests that that's necessarily a permanent thing. I don't think we know enough yet and we'll see as we come out of COVID, but certainly, we've seen merchant rebound considerably and that may well persist. |

Table 4: Detailed descriptions of the annotation process and the justification for choosing a specific rating for each feature for each QA pair.

| Feature | Rating | Sample | Reasoning |
|---|---|---|---|
| Clear | 0 | D | Speaker is nontransparent with their answer, offering a convoluted response which fails to convey a concrete answer. |
| | 1 | B | Speaker clearly states their answer, but their message isn't obvious at a first glance. |
| | 2 | C | Speaker is transparent with their answer, offering a detailed response that conveys their opinion. |
| Assertive | 0 | A | Phrases "think," "would like to see," and "ideally" highlight the speaker's uncertainty regarding the supply chain. |
| | 1 | C | States several facts and speaks certainly about events, but at times also uses phrases such as "I think." |
| | 2 | B | Phrases "no doubt in my mind" and "obviously" highlight the speaker's absolute certainty regarding share gains. |
| Cautious | 0 | B | Clearly states opinion without convolution answer to avoid backlash: "I believe in the federal" and "no doubt in my mind." |
| | 1 | C | Speaker isn't being careful of withholding information, but also isn't making overly bold statements, simply states facts. |
| | 2 | D | Avoids giving concrete information or opinions and speaks in general terms. Avoids giving statements of value (avoids risk). |
| Optimistic | 0 | A | Phrases "still some issues" and "not receiving" suggest a negative outcome regarding the supply chain. |
| | 1 | D | Doesn't utilize words with strong positive or negative connotations. |
| | 2 | B | Phrases "tremendous portfolio" and "took some share" suggest a positive outcome regarding the share gains. |
| Specific | 0 | D | Does not mention any specific ideas or details that relate to the question asked. Answers in a vague and generic context. |
| | 1 | A | Provides surface-level details, "not receiving ... ammunition ... calibers," but no technical information. |
| | 2 | C | Provides specific and technical details, responding with percent revenues, dates, and other data. |
| Relevant | 0 | D | Avoids answering the question, disregarding it and digressing into the effects of COVID on the market. |
| | 1 | C | Addresses some parts of the question well, but fails to respond to others. |
| | 2 | B | Answer elaborates on federal verticals and budget flush, addressing all aspects of the question appropriately. |

Table 5: Details of the various ECTs and the specific asker and responder that we utilized for our sample annotation process wherein the sector labels are defined by Table 7.

| Sample | Ticker | Sector | Asker | Responder | Year | Quarter |
|---|---|---|---|---|---|---|
| A | BGFV | 1 | Sean P. McGowan | Steven G. Miller | 2013 | 2 |
| B | DELL | 0 | Maynard J. Um | Joseph M. Tucci | 2011 | 3 |
| C | DELL | 0 | Benjamin A. Reitzes | Joseph M. Tucci | 2011 | 3 |
| D | EXPE | 6 | Stephen Ju | Peter Kern | 2021 | 2 |

A sample of a annotation process below. The respondent for the answer is Chad Crow, President and CEO of Builders FirstSource in the 2018 Quarter 3 ECT [FirstSource, 2018].

**Question**: And then, just wanted to ask also, with the sharp drop in prices over the last couple of months, is this something where there is an inventory problem inside the industry that's working itself out? Or is this more dislocation from people being worried about whether housing starts keep growing. What do you guys think has caused this sharp decline over the last two months?

**Answer**: To some degree, I almost feel like we're victims of our own actions. I think as prices start to fall, everybody gets a little more cautious on buying. And so, all of a sudden, everybody's gone from, oh, I've got to buy to cover my position because prices are rising, to sitting on the sidelines and waiting to see where things stop falling. And so, in a sense, we all start acting in concert and it ends up  and I think that's part of the problem And I think at some point folks are going to say this thing's hit bottom and everybody is going to start buying again, and we all know what that's going to mean. It's going to mean prices are going to go up. So I think it's a lesser concern about  I mean there's seasonality involved for sure. But I think it's less a concern of the overall health of housing, as it is just people just trying to guess when the bottom is going to be.

After our manual annotation process, we provide each feature a rating of either 0, 1 or 2:

Table 6: Ratings and Justifications for the QA pair.

| Feature | Rating | Justification |
| --- | --- | --- |
| Clear | 2 | The answer has overall clarity in answering the question. It is honest and uses a personal approach, such as an anecdote. Instead of feigning confidence about the future, the speaker transparently discusses market forces and the companys control over price behavior. |
| Assertive | 1 | According to our definition, this answer is not too assertive but rather explanatory. The speaker uses phrases like "I think" and discusses possible cause-and-effect relationships. |
| Cautious | 2 | The answer is very cautious in tone, using words like "think" and "feel." It includes an analysis of potential negative effects, displaying a cautious approach. |
| Optimistic | 0 | The answer has low optimism, as it discusses inflationary pressures within the housing market and heavy speculation. The speaker does not suggest any company strategies to counteract price drops, allowing prices to fluctuate naturally based on consumer behavior. |
| Specific | 2 | The answer is highly specific, closely tailored to the question and the ECT context. It includes a detailed, step-by-step narration of consumer habits and their effect on prices. |
| Relevant | 2 | The answer focuses on the current state of the housing market and remains relevant to the question's overall purpose. The speaker addresses all parts of the question without including extraneous information. |

# H    Dataset Construction and Cleaning

## H.1    Sampling Procedure

Table 7: A generalised industry mapping that better divides the Earnings Calls based on their themes

| Industry | Label |
| --- | --- |
| Business Equipment | 0 |
| Wholesale, Retail, and Some Services (Laundries, Repair Shops) | 1 |
| Consumer Nondurables | 2 |
| Healthcare, Medical Equipment, Drugs | 3 |
| Oil, Gas, Coal Extraction & Products | 4 |
| Manufacturing | 5 |
| Telephone and Television Transmission | 6 |

The original dataset had the following 11 unique values: Business Equipment, Chemicals and Allied Products, Consumer Durables, Consumer Non-Durables, Finance, Healthcare, Medical Equipment, and Drugs, Manufacturing, Oil, Gas, and Coal Extraction and Products, Telephone and Television Transmission, Utilities, Wholesale, Retail, and Some Services (Laundries and Repair Shops).

## H.2    Data Collection Procedure

This selection was balanced across sectors and distributed evenly over time to preclude recency bias. A custom Python script was utilized for the data collection, integrating Selenium for dynamic web page navigation and Beautiful Soup for efficient HTML parsing. This script systematically extracted QA pairs from the Investor Relations sections of company websites. Each earnings call provided a rich source of direct exchanges between company executives and analysts, encapsulating the essence of corporate discourse for subsequent analysis. This raw HTML data was formatted to produce the features: `Asker`, `Respondent`, `Question`, and `Answer` for each QA pair.

## H.3 Data Cleaning

Table 8: Depicting the data cleaning process using Herseys' ECT from 2022 Q3 wherein a QA pair was either retained, omitted or reformatted while creating the unannotated SubjECTive-QA dataset.

| Question | Answer | Omitted, Retained or Reformatted |
|---|---|---|
| I also have a couple of questions. I'm going to start with any comments you might offer about the ammunition supply chain, it's been something that's come up on recent calls. Do we take it from the fact you didn't comment on it that it's gotten a lot better? | Well, I think there's still some issues in terms of the supply chain. I think it has gotten better, but we're still not receiving all the ammunition, all the calibers that we would like to see in the quantities we would – ideally our consumers would like to see them. | Retained |
| Thank you | Great. Great, thanks very much. | Omitted |
| Hi, good morning. | Hi, good morning. | Omitted |
| Okay, thank you. And then, just a follow into that, are you shipping in Holiday product early and would that have been an incremental. But do we see some of that in the third quarter more than we would have seen historically? | Steve Voskuil – Senior Vice President, Chief Financial Officer Yeah, we did. Just like we did with Halloween, started shipping in Holiday a bit early, so you do get some pickup in the third quarter for that. That will take away a little bit from the fourth quarter, maybe on the order of 50 basis points. Michele Buck – Chairman, President and Chief Executive Officer. And we also do that to drive the consumer behavior early as well, just as we did with Halloween, you know as Halloween was still on the floor, Holiday was also out there, so that consumers could also gravitate to the Holiday pretty quickly. A strong Halloween sell-through – yeah, a strong Halloween sell-through really helps us because it helps us get that fast start to Holidays, because it clears the space to be able to put Holiday on the floor. | Reformatted |

# I Feature Selection

For selecting the 6 features PaLM-2 [Anil et al., 2023] was used to aid us with the selection of the features. Each QA pair in the dataset was passed through PaLM-2's API, particularly the text-bison-002. The model parameters set were a temperature of 0.6 to introduce variability in the outputs, alongside a Top-k value of 16, the choices to sample from. The prompt and the code snippet have been provided in 15. This yielded generic features associated with the answer. In order to ensure cohesiveness amongst the properties identified, the model was limited to using Loughran and McDonald Sentiment Word Lists [Loughran and McDonald, 2023]. There were hence a total of 27,470 properties, including overlapping ones. These were then semantically compared using the python library SpaCy to identify similarities across the QA pairs. Once the final list of important properties was generated using SpaCy, the team assessed each property based on our knowledge of the domain as well as the work of related works. Referring to [Huang et al., 2023] and [Chen et al., 2023b], the researchers made a collective decision to choose the properties that were not only the most prevalent within other works within the domain but were also hypothesized to be independent of one another as seen in Figure 4.

# J   Violin Plots

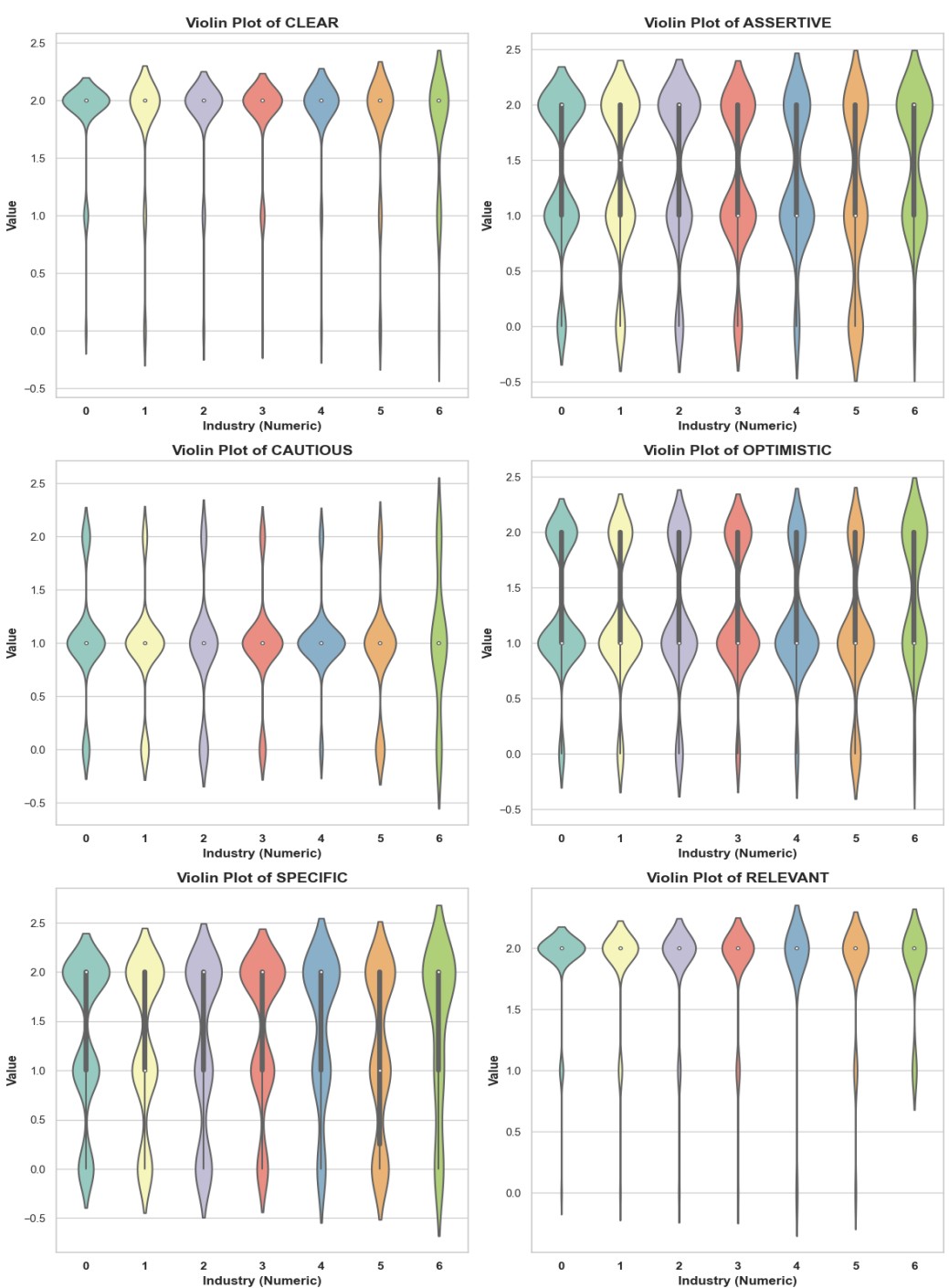

Figure 6: Illustrating all the violin plots across the various industries for the 6 features: (a) `Clear` (b) `Assertive`, (c) `Cautious`, (d) `Optimistic`, (e) `Specific`, (f) `Relevant`

.

# K  Comparison to other Datasets

Table 9: Comparison of SubjECTive-QA with other current industry standard datasets within finance as well as in other domains in terms of the size, number of features, labels and licensing.

| Dataset | Size | Number of Features | List of Labels | License |
|---------|------|-------------------|----------------|---------|
| SubjECTive-QA | 35,711 | 13 | Company Ticker, Question, Answer, Quarter, Year, Asker, Responder, Cautious, Assertive, Optimistic, Specific, Relevant, Clear | CC By 4.0 |
| TAT-QA | 16,552 | 5 | Reasoning, Question, Answer, Scale, Derivation | CC By 4.0 |
| FinQA | 171,000 | 3 | _id, title, text | CC By 4.0 |
| FinArg | 12,623 | 4 | Id, label, start index, end index, text | CC BY-NC 4.0 |
| ConvFinQA | 4,000 | 8 | questions, answers, financial reports, addition, subtraction, multiplication, division, and comparison | MIT |
| MathQA | 37,200 | 7 | Problem, Rationale, options, correct, annotated_formula, linear_formula, category | Apache License 2.0 |
| TruthfulQA | 5,719 | 7 | Type, Category, Question, Best Answer, Correct Answers, Incorrect Answers, Source | Apache License 2.0 |
| Trillion Dollar Words | 12,330 | 5 | index, sentence, year, label, orig_index | CC BY-NC 4.0 |

As evidenced by the table, there are several other datasets that focus on both question answering data as well as other financial sources and some mathematical sources as well. The size of our dataset is comparable to those in the industry. We possess a greater number of features than the average within 9. The comparison of the labels proves to be the most interesting aspect of 9 as the labels within SubjECTive-QA can be used most generally throughout domains aside from finance. The licensing of SubjECTive-QA is CC By 4.0, which is seen as a standard for the industry.

# L    Misinformation Examples

Table 10: Examples of various question-answer pairs from SubjECTive-QA that specifically demonstrate misinformation due to low clarity, specificity and assertiveness.

| Label | Question | Answer |
|-------|----------|--------|
| QA_1 | Yeah. I just wanted to sharpen my pencil here. Could you comment on the outlook for demand in the sector? | Hi, Eric, it's Jean. I just would quibble on the definition of high teens, I would say high teens would probably be closer to 18%, 19% rather than more of a mid teens, 16%. |
| QA_2 | And then as you think about – you mentioned the backlog. Could you elaborate on the specifics? | And keep backlog as – it's a very fluid – it depends a lot on factors we cannot control at this moment. |
| QA_3 | And then, it would be our sense that you felt things are stabilizing. Can you confirm? | Well, I'm having a hard time. As I said earlier, stabilization is happening but slower than expected. |
| QA_4 | Do you think theres a point where you guys could consider different strategies in the near future? | We could consider that, yes, but it's not on the immediate horizon. |

Table 11: QA Features Table

| QA_Label | CLEAR | ASSERTIVE | CAUTIOUS | OPTIMISTIC | SPECIFIC | RELEVANT |
|----------|-------|-----------|----------|------------|----------|----------|
| QA_1 | 0 | 0 | 1 | 1 | 1 | 2 |
| QA_2 | 0 | 0 | 0 | 0 | 0 | 0 |
| QA_3 | 0 | 0 | 1 | 1 | 1 | 2 |
| QA_4 | 0 | 1 | 1 | 1 | 0 | 2 |

As seen from the examples in table 10 such as QA_1 and QA_4, it can be seen that although they mention details about their plans for the future and mention some statistics, they remain ambiguous. Even though they mention specific percentages, they seem to lack confidence in their own statements. Especially within QA_2, we see that they specify that their backlog is fluid and there are a lot of factors but they do not elaborate on the specifics, even though the questioner asks them to. Moreover, this trend follows through in QA_3 as the answerer lacks confidence and are specifically less assertive.

# M    Annotations

## M.1    Annotator Information

Table 12: Detailed background information about each annotator and their background. at the time of annotation.

| Annotator Name | Information |
| --- | --- |
| Siddhant Sukhani | Siddhant Sukhani is an undergraduate Applied Mathematics major with a minor in Computational Data Analysis at Georgia Institute of Technology, Atlanta, Georgia, USA. He is a 20-year-old male from Mumbai, India, and is Indian. His educational background includes the international GCSE curriculum in Mumbai and the International Baccalaureate Diploma program in Jaipur. He was not paid to annotate the ECTs and consents to having his annotations used in this research. |
| Huzaifa Pardawala | Huzaifa Pardawala is an undergraduate Computer Science major at Georgia Institute of Technology, Atlanta, Georgia, USA. He is a 19-year-old male from Mumbai, India, and is Indian. His educational background includes the Indian ICSE and HSC curriculum in Mumbai. He was not paid to annotate the ECTs and consents to having his annotations used in this research. |
| Veer Kejriwal | Veer Kejriwal is an undergraduate Computer Science major with an Economics minor at Georgia Institute of Technology, Atlanta, Georgia, USA. He is a 19-year-old male from Mumbai, India, and is Indian. His educational background includes the international GCSE curriculum and International Baccalaureate Diploma program in Mumbai. He was not paid to annotate the ECTs and consents to having his annotations used in this research. |
| Abhishek Pillai | Abhishek Pillai is an undergraduate Computer Science major at Georgia Institute of Technology, Atlanta, Georgia, USA. He is a 19-year-old male from Mumbai, India, and is Indian. His educational background includes the Indian ICSE and HSC curriculum in Mumbai. He was not paid to annotate the ECTs and consents to having his annotations used in this research. |
| Tarun Mandapati | Tarun Mandapati is an undergraduate Computer Science major at Georgia Institute of Technology, Atlanta, Georgia, USA. He is a 20-year-old male from Frisco, Texas. He was born in India but has completed all of his education in the United States, graduating high school with an International Baccalaureate Diploma. He was not paid to annotate the ECTs and consents to having his annotations used in this research. |
| Rohan Bhasin | Rohan Bhasin is an undergraduate Computer Science major at Georgia Institute of Technology, Atlanta, Georgia, USA. He is a 20-year-old male from Delhi, India, and is Indian. His educational background includes the Indian CBSE curriculum in Delhi. He was not paid to annotate the ECTs and consents to having his annotations used in this research. |
| Andrew DiBasio | Andrew DiBasio is an undergraduate Computer Science major at Georgia Institute of Technology, Atlanta, Georgia, USA. He is a 20-year-old white male from Westford, Massachusetts, and is of Brazilian and European descent. His educational background includes traditional K-12 schooling in the US, from which he obtained a high school diploma. He was not paid to annotate the ECTs and consents to having his annotations used in this research. |
| Dhruv Adha | Dhruv Adha is an undergraduate Computer Science major at Georgia Institute of Technology, Atlanta, Georgia, USA. He is a 19-year-old male from Columbus, Ohio. He was born in India but completed most of his education in America, graduating high school with an International Baccalaureate Diploma. He was not paid to annotate the ECTs and consents to having his annotations used in this research. |
| Chandrasekaran Maruthaiyannan | Chandrasekaran Maruthaiyannan is a graduate Computer Science major with a Machine Learning specialization at Georgia Institute of Technology, Atlanta, Georgia, USA. He is a 37-year-old male from Karur, Tamil Nadu, India, and is Indian. His educational background includes the Anna University and HSC curriculum in Karur. He was not paid to annotate the ECTs and consents to having his annotations used in this research. |

## M.2 Annotator Agreement

Table 13: Detailed Agreement Metrics based on the annotations split up by Feature

| Feature | All Agree | 2 Agree | None Agree |
|---|---|---|---|
| Clear | 1,813 (66.00 %) | 874 (31.82 %) | 60 (2.18%) |
| Assertive | 963 (35.06%) | 1,576 (57.37%) | 208 (7.57%) |
| Cautious | 1,052 (38.30%) | 1,461 (53.19%) | 234 (8.51%) |
| Optimistic | 1,181 (43.00%) | 1,426 (51.91%) | 140 (5.09%) |
| Specific | 1,065 (38.77%) | 1,394 (50.75%) | 288 (10.48%) |
| Relevant | 1,993 (72.55%) | 715 (26.03%) | 39 (1.42%) |

The reason why there is higher inter-annotator agreement for `Clear` and `Relevant` is because most answers given by executives and managers of a company can be easily interpreted as `Clear` and `Relevant` answers, denoting the objectivity of these features. On the other hand, the low inter-annotator agreement for the features `Specific`, `Cautious`, and `Assertive` is due to the subjective nature of the features. Interpreting these three features is dependent on how the annotator interprets answers as `Specific`, `Cautious`, and `Assertive`. Moreover, the executives of a company are bound to give less transparent answers in regards to these three features if there is a setback for the company.

# N    Performance on White House Press Briefings and Gaggles

After obtaining the best hyper-parameters for the best models across the six features, we tested the utility of both our models and our dataset for the transferability. We collected 65 QA pairs from White House Press Briefings and Gaggles [The White House, 2024] to this end and then ran our best models on these QA pairs. The results of the weighted F1 scores can be seen in table 14.

Table 14: Weighted F1 scores of the best performing model for each feature when applied to the White House Press Briefings and Gaggles

| Feature | Weighted F1 Score | Model Used |
|---|---|---|
| Clear | 0.8415 | BERT base model (uncased) |
| Assertive | 0.6947 | RoBERTa base |
| Cautious | 0.3593 | BERT base model (uncased) |
| Optimistic | 0.6432 | RoBERTa base |
| Specific | 0.6992 | FinBERT-tone |
| Relevant | 0.7201 | BERT base model (uncased) |

From this we can infer that in general the White house representatives are inherently more cautious in their terminology when it comes to questions of political nature for diplomacy reasons. Additionally, a high score of `Clear` indicates that speakers must convey information with transparency as to prevent misinformation from spreading.

The weighted F1 scores for the six features indicate that the dataset and the models can thus be generalised and utilised in different fields.

## O Prompts

Table 15: Prompts used for initial feature selection and for benchmarking.

| Prompt Type | Description |
|---|---|
| **Prompt for Feature-Selection (PALM-2)** | Generate exactly 10 features based on the quality and tone of the answer. For example: 'We expect to deal and face with inflation' is a 'Defensive answer' whereas 'We are prepared to take advantage of the inflation' is an 'Aggressive answer'. In this case, you would define an 'Aggression' variable with this answer and any more if needed.A few other examples could be: Clear (vague vs clear) and Optimism (Optimistic vs Non-optimistic).Don't include any justification for the labels. Generate your answer in the following format: 'Here are 10 features based on the quality and tone of an answer: Aggression: The degree to which the answer is assertive or forceful. Clear: The degree to which the answer is easy to understand.Confidence: The degree to which the answer is expressed with certainty.Defensiveness: The degree to which the answer is apologetic or evasive. Optimism: The degree to which the answer is hopeful or positive. Passiveness: The degree to which the answer is meek or submissive. Politeness: The degree to which the answer is respectful or considerate. Relevance: The degree to which the answer is related to the question. Specific: The degree to which the answer is detailed and precise. Tone: The overall emotional quality of the answer.These features can be used to assess the quality of an answer and to identify areas where the answer could be improved. For example, if an answer is unclear, the writer could be asked to provide more detail or to rephrase the answer in a clearer way.If an answer is defensive, the writer could be asked to be more assertive or to provide more evidence to support their claims.' Please note that the examples provided like Aggression, clarity, Confidence, etc. are just examples and for you to understand how to produce the output.You do not need to necessarily give those specific 10 words as an output. The 10 words can be any set of new words. |
| **LLM prompts for Benchmarking** | Given the following feature: {feature} and its corresponding definition: {definition} Give the answer a rating of: 2: If the answer positively demonstrates the chosen feature, with regards to the question. 1: If there is no evident/neutral correlation between the question and the answer for the feature. 0: If the answer negatively correlates to the question on the chosen feature. Provide the rating in the first line and provide a short explanation in the second line. This is the question: {question} and this is the answer: {answer}. |

