# OpenReview forum: "SubjECTive-QA: Measuring Subjectivity in Earnings Call Transcripts' QA Through Six-Dimensional Feature Analysis"
_NeurIPS.cc/2024/Datasets_and_Benchmarks_Track — NeurIPS 2024 Track Datasets and Benchmarks Poster_

### Official Review · Reviewer_eJKB · 2024-07-26

**Rating:** 6
**Confidence:** 3
**Correctness:** Yes.
**Clarity:** Yes.

**Review:**

- The paper addresses a significant gap in NLP datasets by providing manually annotated data for subjective features in long-form QA pairs. This is particularly valuable for financial NLP and has potential applications in other domains.
- The dataset includes annotations from nine annotators across six different subjective dimensions, providing a rich and multi-faceted view of subjectivity in ECTs.
- The authors evaluate a range of models, including both PLMs and LLMs, providing a comprehensive baseline for future work on this dataset.
- The paper demonstrates the dataset's applicability beyond the financial domain by testing on White House Press Briefings, which strengthens its potential impact.
- The work bridges NLP, finance, and misinformation detection, making it potentially impactful across multiple research areas.

**Strengths:**

- While the paper provides benchmarking results, it lacks in-depth analysis of why certain models perform better on specific features. A more detailed error analysis could provide valuable insights for future model development.
- The paper acknowledges that the annotations reflect the specific viewpoints and financial knowledge of the annotators. However, it doesn't provide a detailed discussion on how this potential bias was mitigated or how it might impact the dataset's utility.
- While the six selected features are intuitively relevant, the paper could benefit from a more rigorous justification of why these specific features were chosen and how they relate to existing literature on financial sentiment analysis.
- The paper treats each subjective feature independently, but in reality, these features likely interact in complex ways. An analysis of feature correlations or interactions could provide additional insights.
- The paper doesn't compare SubjECTive-QA to existing datasets in financial sentiment analysis. Such a comparison could better contextualize the contribution of this new dataset.
- Given that the ECTs span from 2007 to 2021, an analysis of how these subjective features might have changed over time could provide valuable insights, especially considering evolving financial regulations and communication strategies.

**Additional Feedback:**

None.

**Documentation:**

Yes.

**Ethics:**

No ethical concerns.

**Limitations:**

Yes.

**Opportunities For Improvement:**

Provided above.

**Relation To Prior Work:**

Please see my review above.

**Summary And Contributions:**

This paper introduces SubjECTive-QA, a novel dataset for analyzing subjective features in long-form question-answer pairs from Earnings Call Transcripts (ECTs). The dataset consists of 2,747 QA pairs manually annotated across six subjective dimensions: Assertive, Cautious, Optimistic, Specific, Clear, and Relevant. The authors benchmark various pre-trained language models (PLMs) and large language models (LLMs) on this dataset, finding that RoBERTa-base performs best overall. They also demonstrate the dataset's generalizability by testing on White House Press Briefings, achieving an average weighted F1 score of 65.97%. The work aims to address the gap in manually annotated datasets for subjective features in formal QA settings, with potential applications in finance, politics, and misinformation detection.

---

> ### Author Rebuttal · Authors · 2024-08-16
>
> We sincerely appreciate your thorough review and the thoughtful insights you've provided. Your feedback has been invaluable in helping us refine our work, and we are grateful for the opportunity to address your comments.
>
> **Error Analysis of Model Performance**
> We acknowledge the discrepancy in performance across features, which is tied to the varying levels of subjectivity in different features. We will include a deeper analysis of this discrepancy within Section 5 of the paper within Figure 3.
>
>
> **Biases in Annotations**
> We have discussed potential biases in detail within our Ethics section, as seen in Appendix A, and addressed these issues in our Annotation Guideline (Section 3.2). We utilized a triple-blind procedure to minimize bias as much as possible.
>
> **Justification of Selected Features**
> The six features were identified using an LLM approach (Section 2, Appendix D, and Appendix H.1) and refined based on their impact and independence, as seen in Figure 4 in Appendix E.1. We believe these features capture the nuances of business communication effectively, particularly in earnings calls as detailed within Appendix D.
>
> **Feature Interactions and Correlations**
> We have included a correlation matrix in Figure 4 in Appendix E.1 and will provide a deeper analysis of this matrix to elucidate the dynamics between pairs or triplets of features. This analysis will improve understanding and demonstrate the usefulness of these metrics.
>
> **Comparison to Prior Datasets**
> We will provide a table comparing SubjECTive-QA to other financial datasets in terms of size, features, and other metrics, highlighting our dataset's unique contributions. The table attached within the one-page PDF will be placed within the camera ready version of the paper, given availability of the extra page.
>
> **Temporal Analysis of Features**
> We have included Figure 2 within the one page PDF depicting the temporal changes of features over time, which will provide valuable insights into the impact of financial regulations and major events on these features. Interestingly, we see the lowest value of the average score of “Assertive” and “Specific” features while “Cautious” had the highest value during the 2008 financial crisis.

---

> ### Comment · Area_Chair_npDg · 2024-08-29
>
> Dear reviewer, The authors have submitted a response to your review. Please ensure that you read it and respond before the end of the discussion period. If appropriate, you may update your scores based on this interaction.

---

### Official Review · Reviewer_nn6k · 2024-07-26

**Rating:** 7
**Confidence:** 5

**Review:**

This paper presents a high-quality and original dataset that addresses a significant gap in the analysis of subjective features in formal QA settings. The dataset is meticulously annotated and benchmarks a range of Pre-trained Language Models, highlighting the importance of domain-specific models in capturing subjective nuances. The clarity of the paper is commendable, with detailed methodology and robust benchmarking.

Pros:
1. Introduces a novel dataset focusing on subjective features in QA settings, filling a crucial gap in the field.
2. The dataset is meticulously annotated by multiple annotators, ensuring high quality and reliability.
3. The paper is well-written with clear explanations of the dataset creation, annotation process, and benchmarking methodology.
4. Provides a valuable resource for researchers, with potential applications beyond the financial domain, including politics, journalism, and sports.

Cons:
1. The benchmark should include more LLMs including GPT-4 and financial LLMs such as fingpt or finma.

**Strengths:**

1. Introduces a novel dataset focusing on subjective features in QA settings, filling a crucial gap in the field.
2. The dataset is meticulously annotated by multiple annotators, ensuring high quality and reliability.
3. The paper is well-written with clear explanations of the dataset creation, annotation process, and benchmarking methodology.
4. Provides a valuable resource for researchers, with potential applications beyond the financial domain, including politics, journalism, and sports.

**Additional Feedback:**

Please see my above reviews.

**Clarity:**

The paper is generally well-written, with clear explanations and a logical structure.

**Correctness:**

Authors should provide more detailed results on the imbalanced performance across features by exploring why certain features like "Clear" and "Relevant" perform better than more subjective ones like "Specific" and "Assertive." This could involve a deeper analysis of the data and model behaviors to identify underlying reasons for these discrepancies

**Documentation:**

It provides a detailed documentation which contains sufficient detail.

**Ethics:**

The authors should detail how they addressed potential biases in annotations, such as providing training to annotators to minimize bias and ensuring diversity among annotators. They should also discuss how these biases might affect the use of the dataset and provide guidelines for mitigating such effects in downstream applications.

**Limitations:**

A more detailed discussion on the ethical implications of deploying models trained on this dataset in real-world scenarios would be beneficial. This includes considering how the models might propagate or mitigate biases and the potential consequences of incorrect subjective interpretations.

**Opportunities For Improvement:**

1. The benchmark should include more LLMs including GPT-4 and financial LLMs such as fingpt or finma.

**Relation To Prior Work:**

Include a table or section comparing "SubjECTive-QA" with key prior datasets and methodologies. Highlight differences in scope, features, annotation processes, and applications. For example, contrast SubjECTive-QA with datasets like FinArg and FinSSLx, emphasizing how it uniquely addresses subjective features in QA settings.

**Summary And Contributions:**

The paper introduces a novel dataset aimed at addressing the subtler forms of misinformation in formal Question-Answer (QA) settings. This dataset, manually annotated by nine annotators, focuses on subjective features across six dimensions. It comprises 2,747 annotated long-form QA pairs from Earnings Call Transcripts (ECTs), where company statements are often subjective and scrutinized for their implications. The study benchmarks the performance of various Pre-trained Language Models (PLMs), revealing that domain-specific models like FinBERT outperform general-purpose ones in capturing subjective nuances. Additionally, the dataset’s generalizability is tested with QA pairs from White House Press Briefings, showcasing its broader applicability beyond the financial domain. The work aims to fill the gap in annotated datasets for subjective features, providing a valuable resource for further research in both financial and other formal QA contexts.

---

> ### Author Rebuttal · Authors · 2024-08-16
>
> We are truly grateful for the time and effort you have invested in reviewing our submission. Your detailed and insightful comments have greatly contributed to our understanding and improvement of the work.
>
> **Benchmarking with Additional Models**
> We have tested on GPT-4o, Llama-3-8B-Chat, and Mixtral-8x7B-Instruct, as shown in Figure 1 of the attached PDF. While we attempted to utilize FinMA, we could not do so. We have provided an anonymous [Google Colab Notebook](https://colab.research.google.com/drive/1ji0i5BzBKbHdB6npph9QQe1cmxHD9iyh?usp=sharing) for the reference. The generated text is the same as the input text and it doesn’t produce any new tokens in the output.
>
> **Ethical Considerations**
> We will update our extensive ethics discussion within Appendix A to discuss the implications of deploying models trained on SubjECTive-QA.
>
> **Imbalanced Performance Across Features**
> The discrepancy in performance is related to the inherent subjectivity of features like "Assertive." This observation aligns with our inter-annotator agreement statistics, which show lower agreement for more subjective features as seen in Table 9 of Appendix F.2. We will move this analysis to the main paper when an additional page is allowed.
>
> **Comparison to Prior Datasets**
> We will provide a table comparing SubjECTive-QA to other financial datasets in terms of size, features, and other metrics, highlighting our dataset's unique contributions. The table attached within the one-page PDF will be placed within the camera-ready version of the paper, given the availability of the extra page.
>
>
> **Biases in Annotations**
> We have addressed potential biases through a comprehensive Annotation guideline and a triple-blind procedure to minimize bias. We will further explore these issues in our Ethics section and provide guidelines for mitigating biases in downstream applications.

---

> ### Comment · Area_Chair_npDg · 2024-08-29
>
> Dear reviewer, The authors have submitted a response to your review. Please ensure that you read it and respond before the end of the discussion period. If appropriate, you may update your scores based on this interaction.

---

### Official Review · Reviewer_xL6k · 2024-07-28
**A QA Dataset for Analyzing Earnings Call Transcripts**

**Rating:** 6
**Confidence:** 3
**Clarity:** The paper is well-written.

**Review:**

This paper addresses the importance of a QA dataset containing public discussions in the financial domain, specifically focusing on Earnings Call QA sessions. The paper is well-written and clearly details the data reproduction process, which ensures the reproducibility of the proposed dataset. However, there are areas for improvement.

**Strengths:**

- The paper introduces fine-grained categories (Assertive, Cautious, Optimistic, Specific, Clear, and Relevant) for analyzing Earnings Call Transcripts. This categorization can be particularly useful in assessing the impact of Earnings Calls on the market.
- The authors provide detailed documentation of the proposed dataset, which is crucial for understanding its quality and ensuring reproducibility.
- The evaluation code and dataset are made publicly available, enhancing reproducibility and facilitating further research in this area.

**Additional Feedback:**

Minor reviews:

In Figure 5 in Appendix, it would be helpful to indicate which industries correspond to the labels 0-6 on the x-axis, either in the figure itself or in the caption. This would improve the readability and interpretation of the data presented.

**Correctness:**

The data construction and experiment design are convincing. However, there are some missing details, as described in "Opportunities for Improvement."

**Documentation:**

Yes. The documentation provided sufficient details.

**Ethics:**

No ethical concerns identified.

**Limitations:**

The authors have addressed the limitations of their work in Section 7.

**Opportunities For Improvement:**

- The proposed dataset provides a valuable financial benchmark. However, there are some areas that could be improved to enhance its overall contribution.
- Title
	- The title could be more descriptive and slightly longer to better reflect the paper's content.
- Clarification of Dataset Applications
	- The authors suggest that the proposed dataset can be used to develop misinformation detection tools (e.g., lines 66-68 in Introduction). However, it's unclear how the dataset, which annotates the tone of answers rather than their truthfulness, can be applied to misinformation detection.
	- Further analysis and explanation would be helpful to understand such applications. The authors could provide concrete examples or case studies demonstrating how the annotated features (Assertive, Cautious, etc.) might correlate with or indicate potential misinformation.
- Details on Experiment Settings:
	- The motivation behind the selection of specific PLMs and LLMs for benchmarking is not clearly explained. For instance, why were large language models with over 70B parameters chosen over smaller LLMs with ~10B parameters? Clarifying these choices would provide valuable context for the experimental design.
- Dataset Size and Transfer Learning Ablations (Section 5.3):
	- The transfer learning ablations use RoBERTa-base fine-tuned on the proposed dataset. However, the dataset size might be limited for training language models, especially for the Assertive category, which shows relatively lower F1 scores in Figure 3.
	- The fine-tuned RoBERTa-based model was tested on 65 QA pairs from White House Press Briefings and Gaggles. It would be beneficial to discuss how these evaluation results indicate the generalizability of the proposed dataset to different fields.

**Relation To Prior Work:**

Yes. The paper summarizes existing datasets and compares them with the proposed dataset.

**Summary And Contributions:**

- This paper introduces a QA dataset for public areas, created from Earnings Call Transcripts. The dataset includes 2,747 manually annotated QA pairs across six features: Assertive, Cautious, Optimistic, Specific, Clear, and Relevant.
- The proposed benchmark is annotated by nine annotators, ensuring a diverse range of perspectives and increasing the reliability of the annotations.
- The dataset is evaluated using both pre-trained language models (RoBERTa, BERT, FinBERT) and large language models (Llama 3 70B Chat, Mixtral-8x22B Instruct).

---

> ### Author Rebuttal · Authors · 2024-08-16
>
> Thank you for your careful consideration of our work and the constructive feedback you have shared. We value your expertise and are committed to integrating your suggestions to enhance the quality of our paper.
>
> **Clarification of Dataset Applications**
> Our dataset focuses on "soft" misinformation, where answers may be factually correct but still misleading. For example, in a financial earnings call, an executive might provide a technically accurate but vague or irrelevant answer, which can mislead resource-constrained retail investors. We will include concrete examples in the camera-ready version of the paper to illustrate this.
>
> **Selection of Large Language Models (LLMs)**
> We initially selected models with over 70B parameters to set a high lower bound for performance as seen within Section 5. However, we have now conducted additional experiments using smaller models, including Llama-3-8B-Chat and Mixtral-8x7B-Instruct, and will include these results in the camera-ready version. We have included these results in Figure 1 of the attached one-page PDF.
>
> **Dataset Size and Transfer Learning Ablations**
> The generalizability of our dataset is supported by the evaluation results on QAs from White House Press Briefings, as detailed in Appendix G. The lower F1 scores for the "Assertive" category suggest that it is more subjective and challenging to model, which aligns with our inter-annotator agreement statistics in Appendix F.2.
>
> **Figure Labels and Readability**
> We have updated the caption for Figure 5 and ensured that numeric codes are clearly labeled in all relevant figures to improve readability and interpretation.

---

> > ### Comment · Reviewer_xL6k · 2024-08-30
> >
> > Thank you for your response. The authors have provided clearer examples of the dataset's application in the financial domain. The additional experiments with smaller language models improve the reproducibility of the proposed dataset for other researchers. I revise my score to 6.

---

> ### Comment · Area_Chair_npDg · 2024-08-29
>
> Dear reviewer, The authors have submitted a response to your review. Please ensure that you read it and respond before the end of the discussion period. If appropriate, you may update your scores based on this interaction. Especially because you have recommended rejection, and the authors have responded to address specific points in your review, it is important that you indicate whether you feel they have addressed any of your concerns.

---

### Official Review · Reviewer_wUf8 · 2024-07-29
**Useful Niche Dataset**

**Rating:** 8
**Confidence:** 4
**Correctness:** The evaluation methods are not proper…

**Review:**

The paper introduces a substantial dataset for a complex task. While the data collection and annotation processes are rigorous, the paper would benefit from enhanced clarity and a more detailed explanation of the labeling schema.

**Strengths:**

- Comprehensive Dataset: The paper contributes a substantial dataset to a previously under-explored domain within Financial Natural Language Processing (FinNLP).
- Rigorous Data Preparation: A detailed methodology for data collection and annotation is presented, ensuring data quality and reliability.

**Additional Feedback:**

None

**Clarity:**

It's well written but assumes a lot about their audience. Do try minimize the use of long, dense sentences

**Documentation:**

The dataset is well-documented

**Ethics:**

No, I don't see any ethical concerns

**Limitations:**

Yes

**Opportunities For Improvement:**

**Clarity**:
The ECT's technical terminology presents a potential barrier to understanding. To improve accessibility, consider dedicating a separate paragraph to contextualizing the research and motivating its significance. Furthermore, certain statements, such as "Benchmarking on our dataset reveals..." require additional explanation. Clarifying the benchmarking methodology (e.g., using LLMs as a classifier) and defining terms like "business subjectivity" and "abnormal returns" will enhance the paper's clarity for a broader audience beyond the FinNLP research community.

**Justification**:
A more detailed explanation of the rationale behind choosing categorical labels would strengthen the paper. Exploring the advantages of categorical labelling over other labelling schemes would provide valuable insights into the research design.

**Title**: The title of the paper does not do it justice. Consider another title

**Relation To Prior Work:**

It's clear enough

**Summary And Contributions:**

The paper introduces SubjECTive-QA, a manually annotated dataset aimed at addressing the challenge of subjective responses in formal Question-Answer settings. Key contributions include:
- SubjECTive-QA dataset: consisting of 2,747 long-form QA pairs from Earnings Call Transcripts (ECTs), annotated by nine annotators.
Annotation Dimensions: Annotation of the dataset across six subjective features: Assertive, Cautious, Optimistic, Specific, Clear, and Relevant.
- Benchmarking and Analysis: Evaluation of Pre-trained Language Models (PLMs), particularly RoBERTa-base and Llama-3-70b-Chat, showing varying performance based on the subjectivity of features.
- Generalizability Testing: Demonstration of the dataset’s broader applicability through testing on QAs from White House Press Briefings and Gaggles, achieving an average weighted F1 score of 65.97%.
- Open Access: Release of SubjECTive-QA under the CC BY 4.0 license, promoting further research and development in subjective QA analysis.

---

> ### Author Rebuttal · Authors · 2024-08-16
>
> We sincerely appreciate the reviewer’s valuable feedback regarding the clarity of our paper. We thank the reviewer for the thoughtful and constructive review, which has greatly helped us enhance our work.
>
> **Clarity of Technical Terminology**
> In the camera-ready version, we will include a dedicated paragraph that contextualizes our research within the broader landscape of Financial NLP, emphasizing its significance and the motivation behind our study if accepted. Expanding on our explanation of the benchmarking methodology, we will provide further details on the use of different models and the reasoning behind our choices. We will also clearly define key terms such as "business subjectivity" and "abnormal returns" to ensure they are comprehensible to readers from various backgrounds.
> To address the reviewer's suggestions, we will incorporate the following text in the revised draft:
> *”abnormal returns (the difference between the actual return of a security and its expected return, generally used to assess the financial impact of specific events.)”*
>
>
> **Justification of Categorical Labels**
> We chose six features—Relevant, Clear, Optimistic, Specific, Cautious, and Assertive—due to their consistent importance across our dataset. This selection was informed by an LLM approach, refined for impact and independence as seen in Appendix D. We opted for categorical labeling due to its clarity and effectiveness in capturing business communication nuances, especially in earnings calls. The three-point scale (0, 1, 2) was chosen for its simplicity and reliability, consistent with established practices in similar domains (e.g., Li et al., 2021). This process is further elaborated in the Annotation Guideline section (Section 3.2). Furthermore, a continuous rating scale would make annotator agreement worse and make the annotation guideline complicated.
>
> **Evaluation Methods**
> We have clarified our dataset quality evaluation, detailed in Section 3 of our paper, and benchmarking methodology, discussed in Section 5. However, a significant amount of the work done was covered within Appendix D, F, H and G  These sections outline our construction, annotation guidelines, and annotator agreement metrics.
>
>
> **Writing Style**
> We have reviewed our writing to minimize long and dense sentences, ensuring the content of our camera-ready paper is accessible to a wide audience if accepted. We truly appreciate your constructive feedback and have incorporated these improvements.

---

> > ### Comment · Reviewer_wUf8 · 2024-08-30
> >
> > Thank you for considering the feedback and making the changes. I look forward to the revised paragraph that will contextualize the research.
> >
> > My original score of 8 remains unchanged, as the 'Opportunities for Improvement' I mentioned did not significantly impact it.

---

> ### Comment · Area_Chair_npDg · 2024-08-29
>
> Dear reviewer, The authors have submitted a response to your review. Please ensure that you read it and respond before the end of the discussion period. If appropriate, you may update your scores based on this interaction. Even though you recommended accept, it's helpful for the authors to know whether their response addressed any of your concerns or suggestions.

---

### Author Rebuttal · Authors · 2024-08-16

We sincerely thank all the reviewers for their time and their thoughtful comments and questions. We are encouraged that the reviewers find that:

 - Our work introduces a novel dataset, SubjECTive-QA, which addresses the challenge of subjective responses in formal Question-Answer settings, particularly within the financial domain (wUf8, xL6k, nn6k, eJKB).
 - We have made significant contributions, including:
   - A substantial novel dataset for a complex task in Financial NLP (FinNLP), annotated with six subjective features: Assertive, Cautious, Optimistic, Specific, Clear, and Relevant (wUf8, xL6k, nn6k, eJKB).
   - Rigorous data preparation and annotation processes, ensuring high quality and reliability of the dataset (wUf8, xL6k, nn6k).
   - A comprehensive benchmarking of various Pre-trained Language Models (PLMs) and Large Language Models (LLMs) on our dataset, with detailed documentation and public availability of both the dataset and evaluation code (xL6k, nn6k, eJKB).
   - Demonstrating the broader applicability of our dataset by testing on White House Press Briefings and showcasing its potential impact beyond finance including politics, sports, journalism, and misinformation detection (wUf8, nn6k, eJKB).
   - The paper is well-written with clear explanations of the dataset creation, annotation process, and benchmarking methodology (wUf8, xL6k, nn6k, eJKB).


**Title Change**: We have updated the title to better reflect the content of our paper, now titled: **"SubjECTive-QA: Measuring Subjectivity in Earnings Call Transcripts’ QA Through Six-Dimensional Feature Analysis"**.

The results for additional models, temporal analysis and additional comparisons to other datasets based on reviewers’ suggestions are included in the attached PDF.

We attempted our best to address the questions as time allowed. We believe the comments and revisions have made the paper stronger and thank all the reviewers for their help. Please find individual responses to your questions below.

---

### Author Response · Authors · 2024-08-23
**Thank You and a Follow-Up to Reviewers**

Dear Reviewers,

Thank you for your positive reviews of our work.

We would appreciate your consideration of our responses to the concerns raised. We hope they address your concerns and aid in your overall assessment of our work.

Thank You,
Authors

---

### Decision · Program_Chairs · 2024-09-26

**Decision:**

Accept (Poster)

**Comment:**

**Overall**: This paper presents a novel human annotated dataset of annotations related to “subjective features” of transcripts that includes features related to tone and clarity of the message. Surprisingly, they find that domain-specific models are a better match for human subjective judgments compared to sota models. The concerns raised by reviewers are mostly addressable in a camera ready revision, or they are suggestions for future work that do not detract from the contributions of the current dataset. On the basis of the rigor applied in dataset creation and the novelty of the benchmark, I believe this paper would be a good fit at NeurIPS.

**Strengths**
- Dataset is comprehensive and well put together (wUf8, eJKB), with clear writing especially around dataset construction (xL6k, nn6k) and annotations (nn6k)
- The dataset is novel (nn6k, eJKB) and authors assess  its generalizability to other domains (eJKB, nn6k, wUf8)

**Weaknesses**
- Lack of error analysis (eJKB). This is, in my opinion, the most serious weakness. The authors commit to adding a more in-depth analysis in the camera-ready, but there is no way for that analysis to be assessed by reviewers, and it’s an important piece in understanding model performance on a fairly niche task.
- Possible limited interest primarily to the FinNLP community (wUf8). However, the reviewer suggests explaining the technical terminology better in the paper, and the paper does demonstrate transfer to tasks relevant outside of finance.
- The title is not a good match for the content of the paper (wUf8, xL6k). Though the reviewers do not make specific suggestions here, I did feel that the title gives the impression that the dataset is much broader or more general than it actually is. The authors indicate their plan to change the title in their rebuttal, and I agree that their proposed new title is more appropriate for the content of the paper.
- The paper does not consider larger domain-specific LLMs (nn6k). The authors provide additional comparisons against sota LLMs in the rebuttal, but state that they were unable to work with domain-specific versions.